# Evaluation of Drug Delivery and Efficacy of Ciprofloxacin-Loaded Povidone Foils and Nanofiber Mats in a Wound-Infection Model Based on Ex Vivo Human Skin

**DOI:** 10.3390/pharmaceutics11100527

**Published:** 2019-10-12

**Authors:** Fiorenza Rancan, Marco Contardi, Jana Jurisch, Ulrike Blume-Peytavi, Annika Vogt, Ilker S. Bayer, Christoph Schaudinn

**Affiliations:** 1Clinical Research Center for Hair and Skin Science, Department of Dermatology and Allergy, Charité—Universitätsmedizin Berlin, Corporate Member of Freie Universität Berlin, Humboldt-Universität zu Berlin, and Berlin Institute of Health, 10117 Berlin, Germany; jana.jurisch@gmx.de (J.J.); ulrike.blume-peytavi@charite.de (U.B.-P.); annika.vogt@charite.de (A.V.); 2Smart Materials, Istituto Italiano di Tecnologia, 16163 Genova, Italy; marco.contardi@iit.it (M.C.); ilker.bayer@iit.it (I.S.B.); 3Advanced Light and Electron Microscopy, ZBS4, Robert Koch Institute, 13353 Berlin, Germany; schaudinnc@rki.de

**Keywords:** wound infection, biofilm, pseudomonas aeruginosa, antimicrobial delivery, polyvinylpyrrolidone, nanofibers

## Abstract

Topical treatment of wound infections is often a challenge due to limited drug availability at the site of infection. Topical drug delivery is an attractive option for reducing systemic side effects, provided that a more selective and sustained local drug delivery is achieved. In this study, a poorly water-soluble antibiotic, ciprofloxacin, was loaded on polyvinylpyrrolidone (PVP)-based foils and nanofiber mats using acetic acid as a solubilizer. Drug delivery kinetics, local toxicity, and antimicrobial activity were tested on an ex vivo wound model based on full-thickness human skin. Wounds of 5 mm in diameter were created on 1.5 × 1.5 cm skin blocks and treated with the investigated materials. While nanofiber mats reached the highest amount of delivered drug after 6 h, foils rapidly achieved a maximum drug concentration and maintained it over 24 h. The treatment had no effect on the overall skin metabolic activity but influenced the wound healing process, as observed using histological analysis. Both delivery systems were efficient in preventing the growth of *Pseudomonas aeruginosa* biofilms in ex vivo human skin. Interestingly, foils loaded with 500 µg of ciprofloxacin accomplished the complete eradication of biofilm infections with 1 × 10^9^ bacteria/wound. We conclude that antimicrobial-loaded resorbable PVP foils and nanofiber mats are promising delivery systems for the prevention or topical treatment of infected wounds.

## 1. Introduction

The number of antibiotic resistant bacteria, as well as the number of immune deficient patients, is increasing. For this reason, infections of post-operative and chronic wounds are becoming a concern for many patients and health care providers [1]. Age-related immune deficiency, diabetes mellitus type 2, venous insufficiency, or immobility are the major conditions leading to chronic wounds [2,3,4]. Chronic wounds are often associated with microbial biofilms, i.e., organized communities of one or more microorganism species encased and shielded by extracellular polymeric substances [5,6]. It was shown that significantly higher concentrations of antibiotics are required to treat biofilm-associated infections [7,8]. One of the reasons for biofilm drug-resistance is the extracellular polymeric matrix, which acts as a shield protecting bacteria from the external environment. In addition, the low concentrations of antibiotics that reach the bacteria in the biofilm favor the formation of persisters, i.e., physiologically inactive dormant cells that are less responsive to antibiotics [9]. Thus, for an efficient treatment of a biofilm-associated infection, a high and sustained concentration of antimicrobial drugs should be achieved at the site of infection and within the biofilm. This is often not fulfilled by most of the available antimicrobial formulations because of unfavorable physicochemical properties of drugs like instability in different biological environments, low solubility, or high molecular weight. New solubilization strategies and innovative pharmaceutical formulations, like polymer conjugates, nanocarriers, and membranes, have the potential to improve drug delivery and thus increase their concentration at the site of infection [10,11].

In this study, we investigated the drug delivery properties of polyvinylpyrrolidone (PVP)-based foils and nanofiber mats. A ciprofloxacin base was chosen as model drug because of its low water-solubility, detectability using fluorescence spectroscopy, and quorum sensing inhibiting properties in *Pseudomonas aeruginosa* (*P. aeruginosa*) at concentrations below the minimal inhibitory concentration (MIC) [12]. It is a broad-spectrum antibiotic that has been approved by the U.S. Food and Drug Administration (FDA) for the treatment of urinary tract and other infections, including skin and skin-structure infections. Nevertheless, it has severe side effects, and in 2016, the FDA recognized the existence of a rare side effect, a potentially permanent syndrome called fluoroquinolone-associated disability [13]. Because of the low water solubility, ciprofloxacin is formulated into tablets for oral use as monohydrochloride monohydrate salt. In this form, its absolute bioavailability is of 60% [14]. Alternatively, a ciprofloxacin base has been formulated in oil-based syrups [15] with a better bioavailability of up to 70% [16]. Formulations for topical use (e.g., ciprofloxacin ophthalmic ointment) are made by dissolving the drug in mineral oil. However, it was shown that drug crystals can still form and even cause the blockage of the bottle nozzle [17]. In general, antibiotics are not used for the topical treatment of chronic wounds because of insufficient drug bioavailability and possible local side effects. However, topical drug delivery would be an advantage in those cases, like an infected diabetic foot, where blood circulation is reduced and systemic therapies are often inefficient. In addition, it has been shown that locally applied antibiotics can reduce the risk of surgical site infections [18]. Finally, topical therapies reduce systemic toxicity and spare the gut microbiome. Thus, a more efficient topical drug delivery would improve both the management and prevention of wound infections.

In a previous study, we used a pH modification strategy to solubilize a ciprofloxacin base and increase the amount loaded on PVP foils and nanofiber mats. Interestingly, the residual acetic acid bound to PVP conferred a peculiar transparency and elasticity to the foils. These delivery systems were shown to be safe and to have a good anti-microbial activity in vitro [19]. In this study, we tested further properties of these materials using a wound infection model based on full-thickness human ex vivo skin. Different models are available to test the antimicrobial activity of a drug. In vitro grown biofilms give the possibility to study drug efficacy toward bacteria grown in communities. Nevertheless, in vitro biofilms are often very different from those found in in vivo infections. In vivo animal models have diverse advantages, including a tissue scaffold, host immune response, and wound healing processes. However, besides ethical reasons, animal studies are expensive and results are not always reproducible in humans. To reduce the number of animal studies, ex vivo models have been proposed for preliminary studies prior to animal studies. Ex vivo human skin has been used for several years to test the skin penetration of chemicals and drugs [20]. Yet, in the last few years, ex vivo porcine or human skin has been used to develop wound infection models [21,22,23,24,25,26,27]. Human full-thickness skin represents not only a three-dimensional scaffold where bacteria can grow, but also a complex environment with extracellular enzymatic activity, antimicrobial peptides, and several different cell populations, including immune-active cells. Thus, human skin infection models are realistic animal-free systems; even if they cannot completely replace in vivo studies, they are useful tools to screen antimicrobial formulations [26].

Using this model, we could measure the drug delivery kinetics, monitor eventual toxic effects, and measure the antimicrobial activity of the tested materials.

## 2. Materials and Methods

### 2.1. Preparation of Ciprofloxacin-Loaded PVP Foils and Nanofiber Mats

Transparent films and nanofiber mats were prepared as described by Contardi et al. [15]. Briefly, films were produced using a solvent casting method starting from aqueous solutions of PVP (3% *w/v*) with a molecular weight (MW) of 360,000 g/mol (Sigma-Aldrich, Milan, Italy), monohydrochloride monohydrate free ciprofloxacin (≥ 98.0% measured by high performance liquid chromatography (HPLC), Sigma-Aldrich, Milan, Italy) and acetic acid (≥ 99.7%, Sigma-Aldrich, Milan, Italy). Three different initial quantities of ciprofloxacin (1.2, 30, and 60 mg) were combined with the polymer and dissolved in acetic acid 30% (*v/v*) to a final volume of 30 mL to reach different drug concentrations (2.2, 44, and 88 mmol). The solutions were cast on Petri dishes (diameter 8.75 cm) for 3 days under an aspirated hood under ambient conditions (16–20 °C and 40–50% r.h.). Then, the films were placed in a vacuum desiccator for 3 more days to complete the removal of excess acetic acid. The nanofiber mats were fabricated by using a vertical electrospinning set-up. Starting solutions of PVP, acetic acid (30% *v/v*), and ciprofloxacin were prepared (final volume 6.2 mL). A higher concentration of the polymer (25% *w/v*) was used with respect to the films to allow for the electrospinning process. Three different concentrations of ciprofloxacin (2.2, 44, and 88 mM) were also prepared for the nanofiber mats. Syringes with a stainless-steel needle (18 gauge) were filled with the three different solutions and connected to a syringe pump (NE-1000, New Era Pump Systems, Inc., New York, NY, USA) working at a constant flow rate (500 μL/h). The needles were clamped to the positive electrode of a high-voltage power supply generating 26 kV at a distance of 24 cm from an aluminum disk used as a collector (diameter of 8.75 cm). Only 2 mL of each solution were electrospun in order to obtain the same amount of ciprofloxacin on both the films and nanofibers.

The morphology of foils and nanofiber mats was analyzed using SEM with a variable pressure JSM-649 microscope (JEOL, Milan, Italy) equipped with a tungsten thermionic electron source working in high vacuum mode and an acceleration voltage of 10 kV. The cross-section of the films was obtained by cutting slices with a UCS ultramicrotome (Leica Microsystems, Wetzlar, Germany) equipped with a glass knife. The specimens were coated with a 10-nm thick film of gold using the sputter coater 208 HR (Cressington Scientific Instruments, Watford, U.K.).

### 2.2. Skin Samples and the Creation of Superficial Wounds

Abdominal skin was obtained after getting informed consent from healthy donors undergoing plastic surgery. The study was conducted according to the Declaration of Helsinki guidelines and after approval by the Ethics Committee of the Charité—Universitätsmedizin Berlin (approval EA1/135/06, renewed on January 2018). Skin explants were used within 2–4 h after surgery. Subcutaneous fat tissue was partially removed, keeping a layer of approximately 5 mm and skin pieces (1.5 × 1.5 cm) were stretched and fixed on a Styrofoam block covered with Parafilm (Bemis Company, Neenah, WI, USA) using needles. The surface of the ex vivo skin (free of injuries or redness) was cleaned with saline solution (0.9% NaCl). The epidermis was then removed with a ball-shaped milling cutter 6 mm in size (No. 28725, Proxxon, Föhren, Germany) mounted on a micro motor handpiece (Marathon N7, TPC Advanced Technology, Inc. Diamond Bar, CA, USA) and rotating at 16,000 rpm. In this way, superficial wounds of approximately 5 mm in diameter were produced [21].

### 2.3. Drug Penetration Kinetics

Using a punch biopsy cutter, discs of 8 mm in diameter were cut out from foils, as well as nanofiber mats containing 44 mmol ciprofloxacin, so that each disc contained approximately 250 µg of ciprofloxacin. Wounds of 5 mm in diameter were produced on 1.5 × 1.5 cm pieces of skin. The disks were applied on the top of the wounds on skin blocs of 1.5 × 1.5 cm that were stretched on a Styrofoam block, placed in a humid chamber, and incubated at 37 °C, 5% CO_2_, and 100% humidity for different time points. Thereafter, non-penetrated material was removed with a cotton swab and the treated wound was removed from the rest of the tissue by means of an 8 mm punch biopsy tool. The tissue was chopped into small pieces and placed in 2-mL tubes filled with HCl (0.1 N, 1.5 mL) to extract the ciprofloxacin. The samples were gently mixed on a shaker for 24 h at room temperature. After centrifugation for 5 min at 300× *g*, the supernatant was collected and placed in triplicate in a 96-well plate (100 µL/well). Ciprofloxacin fluorescence (excitation wavelength: 275 nm, emission wavelength: 480 nm) was measured with an EnSpire^®^ Multimode plate reader (Perkin Elmer, Akron, OH, USA). A standard curve was prepared by dissolving ciprofloxacin in HCl (0.1 N) and preparing dilutions (0.5–10 µg/mL). The amount of penetrated drug was calculated on the basis of the standard curve. Results are presented as the means and standard deviations of three independent experiments.

### 2.4. Metabolic Activity of Skin Cells after the Topical Application of Ciprofloxacin on Ex Vivo Skin Wounds

Wounds were treated with 8 mm discs from PVP-foils loaded with 44 mmol of drug. Wounds treated with NaCl solution (0.9%, 20 µL) served as negative controls, and wounds treated with 50 µg/cm^2^ PVP-coated silver nanoparticles (50 nm size, nanoComposix, San Diego, CA, USA) served as positive controls. Samples and controls were incubated in six-well plates for 20 h in 2 mL RPMI-1640 medium (Gibco, Darmstadt, Germany) without phenol red supplemented with fetal calf serum (FCS) (10%, Gibco, Darmstadt, Germany), glutamine (2mM, Gibco, Darmstadt, Germany), streptomycin (100 μg/mL, Gibco, Darmstadt, Germany), and penicillin (100 I.E./mL, Sigma-Aldrich, Hamburg, Germany). Thereafter, the old medium was replaced with fresh medium added with 500 µL of 2,3-bis-(2-methoxy-4-nitro-5-sulfophenyl)-2H-tetrazolium-5-carboxanilide (XTT) (Roche Diagnostic, Berlin, Germany). After 4 h, the medium from each sample and control was collected (3 × 100 µL) and placed in a 96-well microplate. The optical density at 450 nm was read with the EnSpire^®^ Multimode plate reader using 650 nm as the reference wavelength. Results are presented as a percentage with respect to the values from the control wounds. The averages and standard deviations of values from three independent experiments are reported.

### 2.5. Histological Analysis of Wound Tissue after the Topical Application of Ciprofloxacin-Loaded Foils

Wounds created on ex vivo skin were treated with ciprofloxacin in solution or loaded on foils and nanofiber mats (250 µg/wound). Wounds left untreated served as controls. Skin explant cultures were grown in supplemented RPMI-1640 medium as described in Section 2.4. Every 2 days, the medium (500 mL) was removed and fresh medium (500 mL) was added. After eight days of incubation, the skin was plunge frozen in Tissue Freezing Medium (Leica Microsystems, Wetzlar, Germany) and cryosections were prepared. Hematoxylin and eosin (H&E) staining was performed following the manufacturer’s manual (Roth, Karlsruhe, Germany) and images were taken using optical microscopy with an Olympus IX 50 (OLYMPUS, Hamburg, Germany).

### 2.6. Bacteria Inoculation and Characterization of the PAO1 Wound Infection

An overnight culture of the *P. aeruginosa* strain PAO1 (ATCC 15692) was diluted with tryptic soy broth. The suspension (5 µL, 1 × 10^7^ bacteria) was injected with a 10 µL syringe (26 gauge) with a tapered tip (SGE Analytical Science, Ringwood, VIC, Australia) from the edge of the wound into the dermis. As a control, sterile saline (5 µL, 0.9% NaCl) was injected in an uninfected control wound. Wound samples were incubated in a humidified chamber at 37 °C for 20 h. Biopsies (8 mm) were taken and fixed in a solution of formaldehyde (4%) and glutaraldehyde (0.5% in 50 mmol (4-(2-hydroxyethyl)-1-piperazineethanesulfonic acid) (HEPES) for 48 h at room temperature. Skin samples were then washed in HEPES (50 mmol), and dehydrated in 30, 50, 70, 90, 95, and 100% ethanol. Samples were infiltrated first with a LR White/ethanol solution (1:1, 10 min), and then with pure LR White (2 × 15 min). Successively, samples were transferred to polyallomer centrifuge tubes (5 × 20 mm, Beckman Coulter, Inc., Brea, CA, USA) containing LR White with an accelerator (5 µL/mL monomer). The centrifugation tubes were capped with a gelatin capsule and the samples were left to polymerize for 1 h on ice, and then at 60 °C overnight. Five hundred nanometer sections were cut using a ultramicrotome (EM UC7, Leica, Wetzlar, Germany), mounted on poly-l-lysine slides, and incubated for 10 min on an 80 °C thermo-plate. Sections were stained for 4 min with Richardson’s stain (1% Azure II, 1% Methyleneblue, 1% Borax, Sigma-Aldrich, Hamburg, Germany, then washed with double distilled water (ddH_2_O), and imaged using a microscope (Axiophot, Carl Zeiss Microscopy GmbH, Jena, Germany). For analysis with a scanning electron microscope (SEM), after fixation, one of the sample duplicates was cut with a scalpel in order to reveal the skin profile. The samples were then dehydrated as described above, critical-point dried, mounted on aluminum stubs, sputter-coated with a 12-nm layer of gold-palladium, and finally examined with an SEM (ZEISS 1530 Gemini, Carl Zeiss Microscopy GmbH, Jena Germany) operating at 3 kV using the in-lens electron detector. Images have been cropped, adjusted for optimal brightness, and contrasted using Photoshop Lightroom (version 6.0, Adobe Systems, San Jose, CA, USA).

### 2.7. Antimicrobial Activity of Ciprofloxacin-Loaded Foils and Nanofiber Mats

Each tested setting was done in triplicates with a total of at least three runs. Three different dosages of ciprofloxacin were tested (11, 250, and 500 µg/wound). The treatments with PVP foils and nanofiber mats started either 1 h or 20 h after bacteria inoculation and lasted for 20 h. Thereafter, an 8 mm punch biopsy was used to collect the wound tissue, including some of the surrounding intact skin. The tissue was placed in a 1.5 mL microcentrifuge tube containing saline (0.2 mL) and homogenized for 3 min with a sterile steel pistil mounted on a digital overhead stirrer at 150 rpm (DSL, VELP Scientifica Srl, Usmate, MB, Italy). Thereafter, samples were sonicated for 10 min in an ultrasonic bath (BactoSonic1, Bandelin, Berlin, Germany) at 40 kHz using 200 Weff to detach the bacteria. Volumes of each sample were transferred to the wells of 96-well microplates and diluted in 1:10 steps (20 µL sample + 180 µL saline) by using a multichannel pipette. A volume (5 µL) of each well was plated on square tryptic soy agar plates. After incubation overnight at 37 °C, spotting areas with 5 to 50 colony forming units (CFU) were counted. Mean values of the triplicates were calculated and bacteria number per wound was calculated considering the used dilutions. In the diagrams, bacteria counts/wound are presented as the mean and standard deviation of three independent experiments.

### 2.8. Data Analysis

Data are reported as arithmetic means and standard deviations of at least three experiments. Calculations, data processing, and graphics were prepared with Excel 2018 (Microsoft, Redmond, WA, USA).

## 3. Results and Discussion

### 3.1. Preparation and Characterization of Drug-Loaded Foils and Nanofiber Mats

Starting from aqueous solutions of PVP, acetic acid, and ciprofloxacin at different concentrations, transparent films and nanofiber mats were fabricated using the solvent casting and electrospinning methods, respectively (Figure 1).

The final dry transparent foils had a thickness of 150–180 μm and the absence of crystal formation was verified using SEM (Figure A1, Appendix A). As recently demonstrated, the obtained PVP-based foils represent suitable wound dressings due to their flexibility, adhesion, and resorption properties [19,28]. The nanofiber had an average diameter of 360 ± 80 nm and did not show any defects or beads (Figure A2, Appendix A).

### 3.2. Foils and Nanofiber Mats Had Different Drug Delivery Profiles

Foils and nanofiber mats (discs of 8 mm in diameter) were applied on the top of the 5 mm in diameter wounds created on the 1.5 × 1.5 cm ex vivo skin tissue to provide an applied drug dosage of approximately 250 µg/wound. The skin explants were incubated in humidified chambers in an incubator with 100% humidity. This maintained the wetness of the skin blocks, which favored the dissolution of foils and nanofibers. After different incubation times, the non-penetrated material was removed, 8 mm biopsies were taken, and extracts were prepared to measure the amount of penetrated drug (Figure 2).

Both penetration kinetics could be well-fitted to a second-degree polynomial trend line. Amounts ranging between 12 and 18 µg of ciprofloxacin/wound were detected for foils, whereas amounts ranging between 5 and 10 µg/wound were measured for nanofiber discs. Foils delivered high amounts of ciprofloxacin very quickly (approximately 12 µg after only 2 h), whereas nanofiber mats reached similar concentrations later, after 6 h. Interestingly, foils maintained the reached drug concentration in the wound over 24 h. On the contrary, in samples treated with nanofiber mats, the maximal concentration was maintained for a shorter time range (6–16 h). The decreased amounts of drug in the wound tissue observed for nanofiber mats (Figure 2b) were due to drug diffusion to the neighboring tissue (Figure A3, Appendix B). We suppose that for nanofiber mats, a lower amount of drug penetrated the wound tissue and that, due to diffusion to the nearby skin tissue, a reduction of local drug concentration in the wound was measured after 24 h. On the contrary, foils delivered a higher amount of drug to the wound and could saturate both the wound and the nearby tissue.

In a previous study, the dissolution rate and drug release for both materials was investigated in a phosphate buffer and on a mice full-thickness wound model [19]. Results showed that nanofibers dissolved and released the loaded drug very quickly, within the first two hours, whereas foils took longer. The fast dissolution of nanofibers might result in a high local concentration causing drug precipitation or crystallization, which in turn can hinder drug penetration. Another important aspect to be considered is that the investigated PVP-foils contained higher amounts (1.4%) of residual acetic acid than the nanofiber mats (0.3%) [19], which might also have had an influence on drug solubility and penetration.

In summary, drug penetration experiments using the ex vivo wound model showed that both drug delivery systems could deliver the loaded antibiotic to the wound tissue in a controlled manner. The ability of PVP-foils to maintain a high drug concentration in the wound over 24 h is of significance, especially for the treatment of biofilm infections, which need high antibiotic concentrations to be resolved. On the other side, the more controlled delivery profile of the nanofiber mats might be more useful for the prevention of wound infections.

### 3.3. Local Toxicity of Ciprofloxacin in Full-Thickness Ex Vivo Skin

The wound model was also used to test the potential cytotoxicity of the drug and investigated materials after topical application. For these experiments, tissue blocks were cultured in supplemented RPMI 1640 medium at the air–liquid interface. After 20 h of treatment, tissue viability was tested by means of an XTT test (Figure 3a).

Many cells in skin explants were still metabolically active and an evident formation of the red formazan product was observed. PVP foils did not show any toxicity. This was somehow expected with PVP being a well-tolerated, FDA-approved polymer with many uses, such as a food additive, binder in tablets, and plasma volume expander [29]. Also, foils prepared from a PVP solution in 30% acetic acid resulted in having no effect on skin viability due to the fact that only residual amounts of acetic acid molecules remained within the PVP polymers after solvent evaporation during the preparation process [19]. Finally, the PVP foils loaded with the three concentrations of ciprofloxacin also resulted in having no influence on the overall viability of skin cells after 20 h of incubation. Wounds treated with silver nanoparticles, which release toxic silver ions, served as positive controls. The evident reduction of formazan formation was indicative of silver toxicity toward wound cells. Thus, even if this test gave no information about the type of cells being affected by the tested substances, it was a useful method to detect overall acute toxic effects.

Previous studies have shown that ciprofloxacin has toxic effects on fibroblasts [30] and keratinocytes [31]. Toxicity was shown to be time- and concentration-dependent. For this reason, we tested the effect of ciprofloxacin delivered by the investigated foils with regard to re-epithelialization. After 9 days of incubation in supplemented RPMI-1640 medium, a partial re-epithelialization had occurred in controls, with the re-growing of a keratinocyte layer on the edges and in the wound bed (Figure 3b,d,e). Cell nuclei and the collagen network presented a normal morphology in samples treated with empty PVP foils with residual acetic acid. Only a delayed re-epithelialization was observed, probably due to oxygen deprivation (data not shown). On the contrary, in wounds treated with ciprofloxacin, only a thin epithelial layer was observed. Cells’ nuclei appeared small with an irregular morphology typical of necrotic cells and the collagen matrix was less organized than in the controls (Figure 3c,f,g). This effect was visible in all samples treated with ciprofloxacin at different concentration and was independent of the formulation (data not shown).

In recent years, it has been recognized that fluoroquinolones, despite being well tolerated by a broad portion of patients, can have rare and very disabling side effects like a tendon rupture and irreversible nerve damage [13]. There are several hypotheses on the mechanism of toxicity and on the reason why some persons develop these side effects more than others do. One theory is that there might be a gene variant responsible for a disrupted quinolone metabolism. Our results show that, despite ciprofloxacin inducing no changes in whole skin metabolic activity, it negatively influenced the re-epithelialization of ex vivo wounds. Even if this side effect is tolerable in the case of severe wound infections, these results stress the importance of a more local and controlled release of drugs like ciprofloxacin that have narrow therapeutic windows.

### 3.4. Ciprofloxacin-Loaded Foils and Nanofiber Mats Efficiently Reduced P. aeruginosa Infections

Next, we investigated the antimicrobial efficacy of the investigated drug delivery systems using the ex vivo skin infection model (Figure 4). The PAO1 strain was used, which possesses several proteolytic enzymes, among those collagenases, and was shown to build biofilm-like infections on ex vivo skin [21]. Bacteria inoculated in the wounds grew from 1 × 10^7^ to 1 × 10^9^ bacteria per wound after 20 h of incubation. Macroscopically, the surface of the PAO1-infected wound appeared shiny and yellowish (Figure 4c). Scanning electron microscopy images of the wound surface showed bacteria conglomerates typical of a biofilm. These three-dimensional structures are made of bacteria, extracellular materials, and probably of degraded collagen material (Figure 4d,e). The microscopic pictures of H&E-stained skin sections (Figure 4f) revealed an approximately 10-µm-thick bacteria film on the surface of the wound, but also groups of bacteria deep in the wound tissue (circles). The scanning electron microscopic analysis of the wound profile (Figure 4g,h) confirmed the formation of a bacterial biofilm within the superficial wound tissue, as well as the presence of small agglomerates or scattered bacteria deep in the wound.

In the ex vivo model, most of bacteria grew on the wound’s surface, despite the fact that they were inoculated deep in the connective tissue. *P. aeruginosa* can grow in both aerobic and anaerobic conditions. However, the less favorable conditions and shortage of oxygen in the deeper wound layers resulted in a slower proliferation of bacteria. Nevertheless, these small colonies can be the reason for recurrent infections if not eradicated during a treatment. For this reason, the penetration of adequate drug concentrations deep in the wound tissue and the eradication of all bacterial colonies is a crucial factor for a successful therapy.

*Pseudomonas aeruginosa* has already been cultured on ex vivo skin [24,32,33,34,35]. In this study, we used the ex vivo wound infection model as a three dimensional set up to test the capacity of ciprofloxacin-loaded PVP drug delivery systems to eradicate bacteria located not only on the surface, but also deep in the wound’s tissue. The investigated foils and nanofiber mats were loaded with three different concentrations of ciprofloxacin, corresponding to final doses of 11, 250, and 500 µg per wound. The treatments were applied on the top of the wounds 1 h or 20 h after bacteria inoculation and kept for a further 20 h. After 1 h of inoculation, bacteria were still planktonic and thus less resistant to antibiotics. After 20 h of inoculation, the number of bacteria had increased and bacteria were organized in a biofilm, which was more difficult to treat. Samples treated 1 h after bacteria inoculation served to test the drug efficacy toward a moderate infection with approximately 1 × 10^7^ planktonic bacteria, whereas samples treated 20 h after PAO1 inoculation served to test the efficacy of foils and nanofiber mats on a severe infection with approximately 1 × 10^9^ bacteria primarily organized into a biofilm. The experiment terminated after a total of 40 h.

Macroscopically, a thick, yellow biofilm was visible on the surface of untreated wounds, whereas no bacterial film was visible on the treated samples (Figure 5a).

Drug crystals were visible on the wound edges, where higher concentrations of the drug were reached after the foils or nanofibers had dissolved. When applied 1 h after bacterial inoculation, both foils and nanofiber mats resulted in a complete eradication of the bacterial infection independent from the amount of drug that was applied. This result shows the potential of these drug delivery materials for the prevention of wound infections. In particular, nanofibers should be tailored to obtain a more controlled drug delivery in order to have effective antimicrobial concentrations and limit adverse side effects.

In contrast, the 20-h-biofilm infection was more difficult to treat. PVP foils had the best performance (Figure 5b). Foils loaded with 500 µg ciprofloxacin achieved a complete eradication of the bacteria, and the two lower concentrations resulted in a 3-log reduction of bacteria concentration with respect to the untreated controls. Nanofiber mats had a lower antimicrobial activity with a maximal 3-log reduction achieved by the highest concentration. No effect on bacterial growth was measured for plain PVP foils or foils with residual acetic acid (Figure 5d).

The most striking result was the complete eradication of the biofilm infection achieved with the 500-µg PVP foils. Foils have a higher percentage of residual acetic acid (1.4%) than nanofiber mats (0.1%) [19]. Because acetic acid also has antimicrobial activity, it might have acted synergistically with ciprofloxacin. Nevertheless, the superior efficacy of the foils compared to the nanofiber mats correlated well with the delivery kinetics. Foils delivered higher amounts of the drug and over a longer time than nanofibers did. Phillips et al. assessed the efficacy of several commercially available treatments in a porcine ex vivo wound model and found that time-release silver gel and cadexomer iodine dressings were the most effective in reducing a mature biofilm with a reduction between 5 and 7 logs out of a 7-log total [24]. Using an in vivo mouse model, Roy et al. found that ciprofloxacin-loaded keratin-based hydrogels with a sustained drug release profile could reduce the amount of *P. aeruginosa* in the wound bed by 99.9% without interfering with the key processes of wound healing [36]. Thus, these and our results underline the importance of drug delivery systems with a sustained release profile for the efficient treatment of *P. aeruginosa* wound infections.

## 4. Conclusions

In this study, we used full-thickness ex vivo human skin and a wound infection model to investigate the efficacy and tolerability of ciprofloxacin-loaded PVP foils and nanofiber mats. The model allowed us to test the antimicrobial efficacy of these materials and to correlate it to their delivery properties. Ciprofloxacin was representative of a poorly water-soluble drug to be loaded in PVP-based drug delivery systems. The use of a solubilizer (acetic acid) increased the loading capacity and drug delivery properties of nanofibers and foils. This was in turn a crucial point for the accomplishment of high local drug concentrations that were required for the successful prevention and eradication of the PAO1 biofilm infections.

This strategy can be used to load PVP-based delivery systems with different types of drugs or disinfectants, including high molecular weight moieties. Besides their sustained drug release properties, such matrixes allow for a more precise dosing of the active ingredient in comparison to ointments or creams. Foils and nanofiber mats are flexible and can be easily applied in less accessible skin areas, e.g., the lower back or between the toes, which is an advantage, especially for old and disabled patients. Thus, we conclude that the combination of drug delivery systems and solubilizing agents is a promising strategy to create attractive new pharmaceutical forms for topical drug delivery to treat or prevent wound infections.

## Figures and Tables

**Figure 1 pharmaceutics-11-00527-f001:**
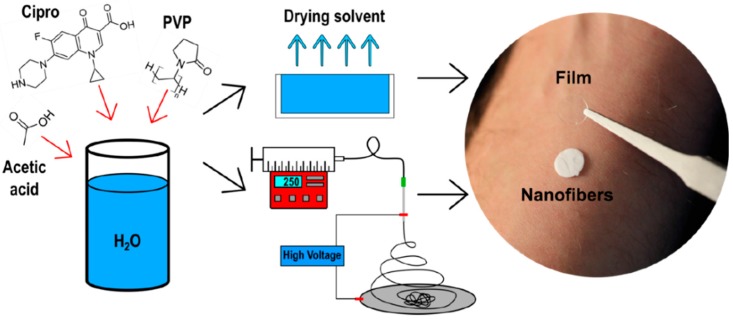
Schematic representation of the methods used for the film and nanofiber mat fabrication, and pictures of the discs used in the experimental set up. Cipro—ciprofloxacin; PVP—polyvinylpyrrolidone.

**Figure 2 pharmaceutics-11-00527-f002:**
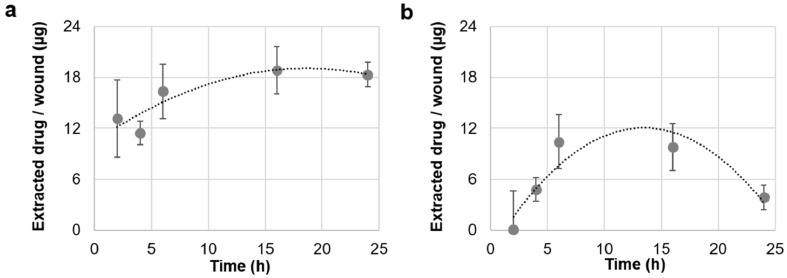
Time-dependent concentration of ciprofloxacin into wound tissue after the topical application of ciprofloxacin-loaded foils (**a**) and nanofibers (**b**). Ciprofloxacin in wound tissue extracts was measured by means of a fluorescence microplate reader and drug concentration was calculated from standard curves. Means and standard deviations from three independent experiments using skin from three different donors are reported.

**Figure 3 pharmaceutics-11-00527-f003:**
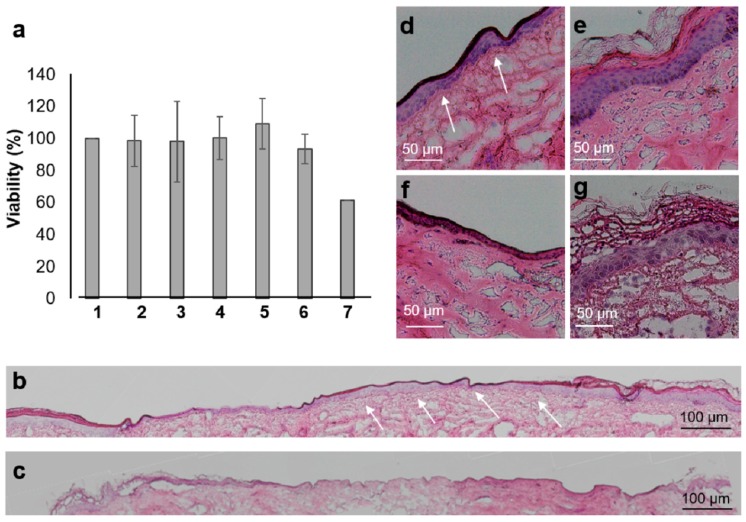
Local toxicity of ciprofloxacin delivered using PVP-foils. (**a**) The XTT assay was run with a skin biopsy previously treated for 20 h with: 0.9% NaCl (1); PVP foils prepared from solutions in ddH_2_O (2); PVP foils prepared from solutions in 30% acetic acid (3); PVP foils loaded with 11 µg (4), 250 µg (5), and 500 µg (6) of ciprofloxacin; and 50 µg of PVP-coated silver nanoparticles (7). Means and standard deviations from three independent experiments are reported. (**b**,**c**) Picture collage of H&E stained sections after 9 days of tissue culture of (**b**) an untreated wound and (**c**) a wound treated with a PVP foil loaded with 250 µg ciprofloxacin. (**d–g**) Details from the centre (**d**,**f**) and the edges (**e**,**g**) of control (**d**,**e**) and treated (**f**,**g**) wounds. Arrows show a newly formed epithelial layer.

**Figure 4 pharmaceutics-11-00527-f004:**
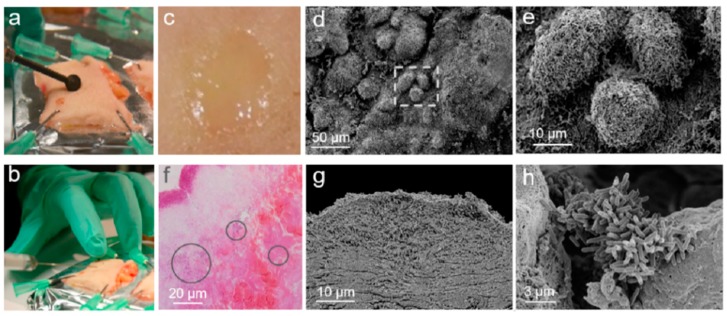
Wound infection model with the *P. aeruginosa* strain PAO1. (**a**) A superficial wound of approximately 5 mm in diameter was produced on skin explants. (**b**) Bacteria (1 × 10^7^ per wound) were inoculated and skin explants, which were incubated for 20 h in the previous analysis. (**c**) Microscopic appearance of a representative wound 20 h after the PAO1 injection. (**d**,**e**) SEM images of the wound surface at two different magnifications showing the typical morphology of bacterial communities in the biofilm. (**f**) Wound section stained with H&E showing a bacteria layer on the surface of the wound and bacteria agglomerates deep into the tissue (circles). (**g**,**h**) SEM images of wound sections at different magnifications confirming bacterial growth on the surface (**g**) and deep in the wound tissue (**h**).

**Figure 5 pharmaceutics-11-00527-f005:**
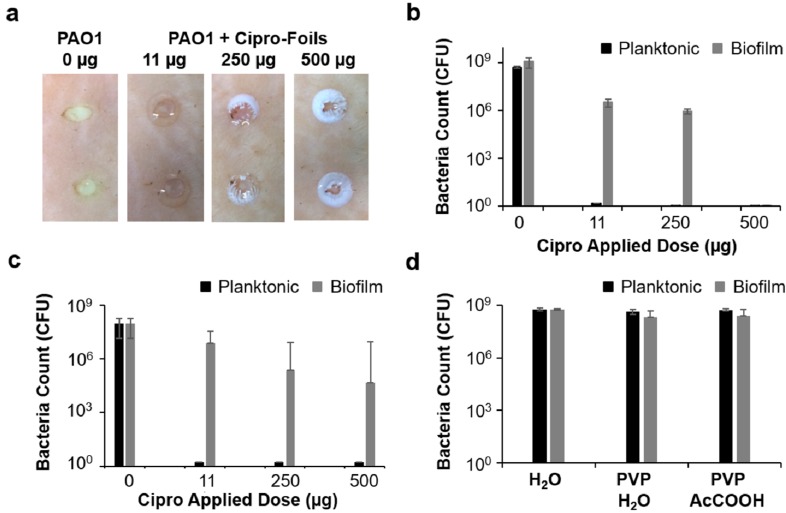
Antimicrobial efficacy of PVP foils and nanofiber mats with different ciprofloxacin payloads applied on wounds infected with planktonic bacteria in a biofilm. (**a**) Representative pictures of untreated and treated wounds after 40 h of incubation. (**b**,**c**) Bacteria count in tissue extracts of wounds treated with foils (**b**) or nanofiber mats (**c**). (**d**) Bacteria counts in control wounds treated with plain PVP foils prepared from solutions in water or 30% acetic acid. Means and standard deviations of three independent experiments (three donors) run in triplicate are reported. Cipro—ciprofloxacin; AcCOOH—acetic acid.

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
