# Peer review of "Evaluation of Drug Delivery and Efficacy of Ciprofloxacin-Loaded Povidone Foils and Nanofiber Mats in a Wound-Infection Model Based on Ex Vivo Human Skin"

_pharmaceutics, 2019, doi:10.3390/pharmaceutics11100527_

Round 1

Reviewer 1 Report

I have provided my comments and suggestions below. To maintain the high quality of studies published in Pharmaceutics, I believe that the authors need to address these comments in the manuscript before considering the it for publication. 

Page 3, line 99: What is water: acetic acid ratio in the final solution? Page 6, Figure 2: The Y scale (ug/wound) of both a & b are not clear. Cleary state much/what size of wound it is? As the error bars are huge and have overlap in panel a, how the authors justify the successful penetration of drugs into the wound? In panel b, why drug penetration was decreased after 25 h? Almost a similar trend is observed in Figure B1. Clearly state your justifications on this issue and in vitro drug release test results must be provided as well. Page 6, line 255: determine the residual concentration of Acetic acid in both foils and nanofiber mats. Figure 3, Caption: where are arrows in the figures? Justify your finding presented in Figure 5. Why after 20 h of treatment, a remarkable number of CFU can be observed?

Author Response

We are thankful for your comments that helped us to improve the quality of our manuscript.

Below the answers to your questions (the mentioned line numbers refer to the highlighted version of the manuscript):

“Page 3, line 99: What is water: acetic acid ratio in the final solution?”

Nine mL of glacial acetic acid were added to 21 mL of water and used to dissolve the polymer as well as the drug. Thus, the final acetic acid / water ratio was 1:2.3, corresponding to 30% (v/v) acetic acid. The information was added at line 102.

Page 6, Figure 2: The Y scale (ug/wound) of both a & b are not clear.

The y-axis reports the amount of drug extracted from wound tissue at the end of the incubation time. An 8 mm punch biopsy was taken, drug was extracted and quantified by means of standard curves. The title of y-axis was changed in “Extracted drug / wound [µg]”.

Cleary state much/what size of wound it is?

Superficial wounds were created where epidermis was completely removed. They have a size of 5 mm in diameter, as stated in Material and Method, line 128. The information was added at lines 22-23 and at line 139.

As the error bars are huge and have overlap in panel a, how the authors justify the successful penetration of drugs into the wound?

The error bars are big because drug penetration was studied using skin tissue from three different donors. Skin tissue varies strongly depending on age, sex, nutrition, etc. This can influence the environment of the wound and the penetration of drug. However, the penetration curves for each single experiment/donor had always the same shape, as shown in the figure below.

We also measured drug in the neighboring tissue (Fig. B 1), which is mostly due to diffusion of the drug penetrated in the wound and only in part to drug penetrated across the neighboring intact skin. Thus, we assume that the released drug penetrated in the wound bed and then diffuses to the rest of the tissue.

In panel b, why drug penetration was decreased after 25 h? Almost a similar trend is observed in Figure B1. Clearly state your justifications on this issue.

After penetration in the wound bed, drug can diffuse to the neighboring tissue as shown in Fig. B1. In the case of nanofiber mats, the amount of drug measured in wound tissue after 24 hours had decreased because after a maximum concentration was reached (6-16h) the penetrated drug had distributed to the rest of the tissue. This explanation was added in the text (lines 252-257)

In vitro drug release test results must be provided as well.

An in vitro release study has been done in a previous work cited as reference 19 (Contardi et. al Transparent ciprofloxacin-povidone antibiotic films and nanofiber mats as potential skin and wound care dressings. Eur. J. Pharm. Sci. 2017, 104, 133-144).

This was added in lines 260-261

Page 6, line 255: determine the residual concentration of Acetic acid in both foils and nanofiber mats.

This was already determined in the previous work (Ref. 19). The information was added in the text at lines 265-266.

Figure 3, Caption: where are arrows in the figures?

By mistake an old version of Fig. 3 was used, The newest version of the Figure with arrows has been inserted.

Justify your finding presented in Figure 5. Why after 20 h of treatment, a remarkable number of CFU can be observed?

The legend given in Fig. 5 (1h and 20h) refers to the time between bacteria inoculation and begin of treatment. After 1h of inoculation, bacteria are still planktonic and, thus, less resistant to antibiotics. After 20h of inoculation, the number of bacteria has increased and bacteria have organized in a biofilm, which is more difficult to be treated. For example, Wang et al. showed that the concentration of ciprofloxacin required to treat biofilm is 128-time higher than that needed to treat planktonic bacteria (Wang H, Wu H, Song Z, Høiby N. Ciprofloxacin shows concentration-dependent killing of Pseudomonas aeruginosa biofilm in vitro. Journal of Cystic Fibrosis. 2010; 9:S41).

This explanation has been added at lines 358-360.

To avoid misunderstanding the legend in Fig. 5 was changed.

Reviewer 2 Report

Dear Authors,

Overall, manuscript is good. However, Please discuss or explain whether there will be any residual acetic acid in formulation and its impact on the skin.

Author Response

We are thankful for the comments to our manuscript.

Concerning your comment on residual acetic acid, this was measured by means of nuclear magnetic resonance (NMR) in both foils and nanofiber mats as reported in Reference 19 (Contardi et. al Transparent ciprofloxacin-povidone antibiotic films and nanofiber mats as potential skin and wound care dressings. Eur. J. Pharm. Sci. 2017, 104, 133-144). The amounts of acetic acid were very low (1.4% for foils and 0.3% for the nanofiber mats). The information was added at lines 265-266. No impact on skin metabolism (Fig. 3a) or altered skin morphology were observed. Comments on this were added at lines 305-307.

Reviewer 3 Report

Authors introduce examination of their previously published mixture of ciprofloxacin loaded on polyvinylpyrrolidone, using human ex vivo skin. Overall, the study was well designed and well described. There are few concerns that must be addressed;

1- In the abstract, line 26, authors wrote: "The delivered drug had no effects on the overall skin metabolic activity but delayed the wound healing process as observed by histological analysis."

Please check if the term "delayed" is the right description of results.

2- Authors previously introduced this mixture in "Transparent ciprofloxacin-povidone antibiotic films and nanofiber mats as potential skin and wound care dressings." and tested it on mice. Therefore, the novelty of this manuscript is on its application on human skin. 

However, the wound-infection model of human skin is not well described in the current study. Details on how the wounds were created are not well described. 

Therefore, we recommend authors to describe in more detail regarding their wound infection human skin model. Providing illustrations of the process and the system would also help readers to better understand the concept.

3- The literature review on the comparison of the wound healing process compared with the system utilized in this study, would help readers to be able to predict the effectiveness of the mixture in actual clinical application.

Also, it would be beneficial if authors include a comparison of the mixture permeation rate compared with in vivo. (through literature review) 

4- Although the authors mentioned applying their mixture on animals is costly and requires detailed examinations, there are animals like a guinea pig that pose a closer skin structure as human and would provide a better evaluation of their mixture in vivo. 

Author Response

We are thankful for the valuable comments that helped us to improve the quality of our manuscript.

Below the answers to your questions (the mentioned line numbers refer to the highlighted version of the manuscript):

1- In the abstract, line 26, authors wrote: "The delivered drug had no effects on the overall skin metabolic activity but delayed the wound healing process as observed by histological analysis."

Please check if the term "delayed" is the right description of results.

 We agree that the word “delayed” is not appropriate and changed it with “influenced”.

2- Authors previously introduced this mixture in "Transparent ciprofloxacin-povidone antibiotic films and nanofiber mats as potential skin and wound care dressings." and tested it on mice. Therefore, the novelty of this manuscript is on its application on human skin. 

However, the wound-infection model of human skin is not well described in the current study. Details on how the wounds were created are not well described. 

Therefore, we recommend authors to describe in more detail regarding their wound infection human skin model. Providing illustrations of the process and the system would also help readers to better understand the concept.

 The wound-infection model has been described more in details and two new pictures have been added in Figure 4.

3- The literature review on the comparison of the wound healing process compared with the system utilized in this study, would help readers to be able to predict the effectiveness of the mixture in actual clinical application.

Also, it would be beneficial if authors include a comparison of the mixture permeation rate compared with in vivo. (through literature review) 

In this study, we did not aim at investigating the wound healing process. Besides measuring drug delivery kinetics and antimicrobial activity, the third aim of the study was to evaluate the safety of the used drug delivery systems. We just performed a histological study without investigating parameters for the evaluation of wound healing, like expression of different keratins, filaggrin, or tight junction proteins. For this reason, we think that the reported results are not enough for a comparison with in vivo would healing studies.

To our knowledge there are no publications in literature that report on in vivo penetration of ciprofloxacin in wounds. We only found penetration studies across corneal tissue (Cantor LB, et al. Ocular penetration of levofloxacin, ofloxacin and ciprofloxacin in eyes with functioning filtering blebs: investigator masked, randomised clinical trial. Br J Ophthalmol. 2008; 92(3):345-7) and (Healy DP et al. Concentrations of levofloxacin, ofloxacin, and ciprofloxacin in human corneal stromal tissue and aqueous humor after topical administration. Cornea. 2004 Apr;23(3):255-63) or drug penetration in wound tissue after systemic administration and negative pressure treatment (Rowan MP et al. Wound Penetration of Cefazolin, Ciprofloxacin, Piperacillin, Tazobactam, and Vancomycin During Negative Pressure Wound Therapy. Adv Wound Care (New Rochelle). 2017;6(2):55-62). All in vivo studies investigated drug antibacterial efficacy and wound healing but did not evaluate drug concentrations in wound tissue.

4- Although the authors mentioned applying their mixture on animals is costly and requires detailed examinations, there are animals like a guinea pig that pose a closer skin structure as human and would provide a better evaluation of their mixture in vivo. 

We agree with the fact that animal studies are still needed to evaluate in vivo efficacy of new antimicrobial formulations. However, the developed ex vivo wound model gives the possibility of reducing the number of animal studies by performing preliminary screening experiments. This concept has been written at lines 84-85.

Round 2

Reviewer 1 Report

The authors provided a comprehensive response to the comments. I accept this manuscript for publication in Pharmaceutics.